# Which SDM Model, CLIMEX vs. MaxEnt, Best Forecasts *Aeolesthes sarta* Distribution at a Global Scale under Climate Change Scenarios?

**DOI:** 10.3390/insects15050324

**Published:** 2024-05-01

**Authors:** Umer Hayat, Juan Shi, Zhuojin Wu, Muhammad Rizwan, Muhammad Sajjad Haider

**Affiliations:** 1Sino-France Joint Laboratory for Invasive Forest Pests in Eurasia, Beijing Forestry University, Beijing 100083, China; oomarcassi6116@gmail.com (U.H.);; 2Beijing Key Laboratory for Forest Pest Control, School of Forestry, Beijing Forestry University, Beijing 100083, China; 3Department of Plant Medicine (Entomology), College of Agriculture and Life Science, Kyungpook National University, Daegu 41566, Republic of Korea; 4Department of Forestry, College of Agriculture, University of Sargodha, Sargodha 40100, Pakistan

**Keywords:** pest risk analysis, *Aeolesthes sarta*, species distribution modeling, CLIMEX, MaxEnt, insect pest, quarantine pest, climate change

## Abstract

**Simple Summary:**

Our study evaluates the global risk of *Aeolesthes sarta* establishment using mechanistic and correlational niche models. The pest, known to affect various hardwood trees, poses a threat to international trade due to its quarantine status. The CLIMEX model, based on species-specific physiological thresholds, and MaxEnt model, using species occurrences and climatic data, project its potential distribution. Both models align well with the current distribution, predicting broader ranges in the central and southern hemispheres, excluding extreme northern regions. Future climate changes may expand its range, particularly in Europe and North America, where its host species are present. Temperature and precipitation are key factors influencing its distribution. These models offer valuable insights for policymakers and trade negotiators to make science-based decisions on pest management and trade agreements, aiding in monitoring potential pest introductions globally.

**Abstract:**

A precise evaluation of the risk of establishing insect pests is essential for national plant protection organizations. This accuracy is crucial in negotiating international trade agreements for forestry-related commodities, which have the potential to carry pests and lead to unintended introductions in the importing countries. In our study, we employed both mechanistic and correlative niche models to assess and map the global patterns of potential establishment for *Aeolesthes sarta* under current and future climates. This insect is a significant pest affecting tree species of the genus *Populus*, *Salix*, *Acer*, *Malus*, *Juglans*, and other hardwood trees. Notably, it is also categorized as a quarantine pest in countries where it is not currently present. The mechanistic model, CLIMEX, was calibrated using species-specific physiological tolerance thresholds, providing a detailed understanding of the environmental factors influencing the species. In contrast, the correlative model, maximum entropy (MaxEnt), utilized species occurrences and spatial climatic data, offering insights into the species’ distribution based on observed data and environmental conditions. The projected potential distribution from CLIMEX and MaxEnt models aligns well with the currently known distribution of *A. sarta*. CLIMEX predicts a broader global distribution than MaxEnt, indicating that most central and southern hemispheres are suitable for its distribution, excluding the extreme northern hemisphere, central African countries, and the northern part of Australia. Both models accurately predict the known distribution of *A. sarta* in the Asian continent, and their projections suggest a slight overall increase in the global distribution range of *A. sarta* with future changes in climate temperature, majorly concentrating in the central and northern hemispheres. Furthermore, the models anticipate suitable conditions in Europe and North America, where *A. sarta* currently does not occur but where its preferred host species, *Populus alba*, is present. The main environmental variables associated with the distribution of *A. sarta* at a global level were the average annual temperature and precipitation rate. The predictive models developed in this study offer insights into the global risk of *A. sarta* establishment and can be valuable for monitoring potential pest introductions in different countries. Additionally, policymakers and trade negotiators can utilize these models to make science-based decisions regarding pest management and international trade agreements.

## 1. Introduction

Climate change has resulted in severe weather occurrences like droughts and floods, exerting extensive impacts on the global ecosystem [1]. These effects include escalating sea levels [2], shifts in crop production regions [3,4], and alterations in species distribution [5,6,7]. As per the Intergovernmental Panel on Climate Change (IPCC), Earth’s temperature is anticipated to increase by approximately 1.4 to 5.8 degrees Celsius between 1990 and 2100. Moreover, precipitation levels are forecasted to rise to 1.0% in mid- and high-latitude areas and 0.3% in tropical zones [8]. The shifts in climate have significantly changed species phenology, biodiversity, potential distribution range, and habitats [1]. These changes have also led to the invasion of exotic species and the extension of the growing period, primarily attributed to temperature increases [2,6]. Climate, recognized as the paramount factor governing growth and development [9], heavily impacts pests due to climate change [4]. Consequently, pests can expand their range, resulting in heightened damage to human livelihoods [1].

*Aeolesthes sarta* (syn. *Trirachys sartus*), commonly known as Sarta longhorned beetle (SLB), is a prominent member of the Cerambycidae family [10]. This polyphagous beetle species primarily targets broadleaved tree species from the genera *Populus*, *Juglans*, *Acer*, *Salix*, *Malus*, *Platanus*, and *Ulmus* [10]. The larvae of the *A. sarta* feed internally on both live and dead plant tissues, resulting in structural damage to host trees and obstructing the flow of nutrients and water [11]. Consequently, this damage leads to the demise of multiple branches and, ultimately, the entire tree [10,12,13]. *Aeolesthes sarta* is believed to have originated in Pakistan and the western regions of India, extending its significant distribution into Afghanistan, Iran, and other Central Asian countries [14,15]. It can thrive in forests up to 2000 m above sea level [10,16]. The region’s warm temperatures and the presence of preferred host tree species create an optimal environment for its proliferation [4]. This species poses significant challenges, particularly in hot and arid climates [17].

In Central Asia, the longhorn beetle emerges as a significant pest, causing substantial damage to various broadleaved trees [10]. *Populus alba* (from genus *Populus*) stands out as a primary host species repeatedly targeted and harmed by *A. sarta* within its native territories [10,18,19,20,21,22]. *A. sarta* exhibits a distinctive ability to attack and reproduce on the main stem and large branches [19]. Infestations are particularly noticeable in highland forests, leading to declines in poplar tree populations. Despite being a vital wood source for industries [21], poplar trees face destruction by the *A. sarta* borer, rendering them unsuitable for industrial purposes [20]. In the Mustang, Balochistan area, a survey of 200 *P. alba* trees revealed severe infestation in 100%, with 34% being destroyed [20]. *Populus alba* appears as a dominant host species afflicted by *A. sarta* in the Mashhad and Zahedan regions of Iran, with infestation rates reaching 100% and mortality rates recorded at 49% and 11%, respectively [19]. 

*A. sarta* demonstrates a broad feeding range, targeting 15 distinct types of trees, thereby establishing itself as one of India’s most prominent pests of hardwood tree species, and impact extends across natural and artificial forest stands [23]. It stands out as one of the most destructive pests affecting walnut trees (*Juglans regia*) in India [24]. Furthermore, *A. sarta* attacks have caused substantial economic losses in Turkmenistan in apple orchards and shelter belts [25,26]. Certainly, in Tashauz, Turkmenistan, this beetle has been responsible for the demise of many tall trees at urban sites [27]. Its infestations have become pervasive in Tajikistan’s Beshkent and Vakhsh Valleys, with few trees remaining unaffected [28]. In Pakistan, similar to numerous other broadleaved tree species, including genera like *Populus*, *Salix*, *Acer*, and *Platanus*, among others, shelter belts and apple orchards have been notably infested by the *A. sarta* [10,18,29]. This beetle is identified as one of Pakistan’s lethal poplar borers [20], causing substantial harm to numerous *Populus* plantations across the country [20,21,22].

Assessing the potential geographical distribution of a species primarily relies on species distribution models (SDMs) [30]. These models encompass mechanistic approaches like CLIMEX and correlative methods like MaxEnt [31]. CLIMEX focuses on the species’ ecophysiological response to its environmental niche, while MaxEnt emphasizes the statistical relationship between species occurrence sites and environmental variables [32]. CLIMEX, grounded in physiological parameters and biological traits, aims to elucidate the climate impact on invasive species [33,34,35]. MaxEnt, a machine learning algorithm, predicts the species’ probability distribution with maximum entropy while considering environmental constraints [36]. These models are commonly used for species damaging to agriculture [37,38,39]. Relying solely on one model may expose vulnerabilities to the limitations of invasive species characteristics, algorithmic applications, and data availability [31]. Combining mechanistic and correlative models has been shown to yield more stable and accurate outcomes [31,32,40].

Choosing *A. sarta* as a subject species for this research holds significant importance for several key reasons: (1) *A. sarta* stands out for its profound ecological and economic ramifications as a forest pest, impacting various tree species across diverse regions [10]; (2) given its sensitivity to climate variables, *A. sarta*’s distribution is poised to undergo significant shifts due to climate change, rendering it a pivotal focus for predictive modeling [6]; and (3) delving into the potential distributional changes of *A. sarta* under climate change scenarios holds promise for informing strategic pest management approaches, thereby mitigating its adverse effects on forest ecosystems and human livelihoods. Accordingly, we have constructed four possible research questions for this study: (a) which SDM model, CLIMEX or MaxEnt, provides more accurate predictions of *A. sarta* distribution under current climatic conditions? (b) How do the predictions of *A. sarta* distribution differ between the CLIMEX and MaxEnt models when considering future climate change scenarios? (c) What are the key environmental variables driving the distribution of *A. sarta* according to each SDM model? (d) How do the uncertainty levels vary between the predictions of CLIMEX and MaxEnt models for *A. sarta* distribution under climate change scenarios? This study uses historical occurrence data to compare CLIMEX and MaxEnt models in predicting *A. sarta*’s current distribution. It evaluates the models’ sensitivity to climate change scenarios, identifies significant environmental variables, and validates predictions against independent datasets. Additionally, it examines uncertainties’ impact and forecasts population shifts under varied climate scenarios, informing pest management strategies.

## 2. Materials and Methods

### 2.1. Aeolesthes sarta Distribution Datasets

Data on the global distribution of *A. sarta* before 3 March 2024 were collected from various reputable sources, including the Centre for Agriculture and Bioscience International (CABI → https://www.cabi.org/ (accessed on 3 March 2024)), the Global Biodiversity Information Facility (GBIF → https://www.gbif.org/ (accessed on 3 March 2024)), European and Mediterranean Plant Protection Organization (EPPO → https://gd.eppo.int/ (accessed on 3 March 2024)), scientific databases like Science Direct and Web of Science, as well as the published literature. In cases where published research lacked precise geographic co-ordinates for positive sites, co-ordinates were determined using Google Earth (https://www.google.com/earth/, accessed on 4 March 2024) and Google Maps (https://www.google.com/maps/preview, accessed on 4 March 2024). Duplicate occurrences were excluded from the dataset. To mitigate sampling bias and spatial autocorrelation in the model, we employed the ENMTools 1.0 software (https://github.com/danlwarren/ENMTools accessed on 10 March 2024) and Species Distribution Model (SDM) Toolbox v2.5 of ArcGIS 10.7. Occurrence data underwent spatial rarefaction, setting a minimum distance of 30 km between each record to enhance model quality [41,42]. Out of 51 known occurrences, 48 were finalized to run both models.

### 2.2. Climatic Data

The meteorological data utilized for the CLIMEX model were sourced from CliMond (https://www.climond.org/ accessed on 10 March 2024), encompassing a spatial resolution of 10 min and spanning the period from 1960 to 1990 [43]. These data, formatted for CLIMEX interpretation, comprised comprehensive monthly averages of minimum and maximum temperatures, precipitation, and relative humidity values at 9 a.m. and 3 p.m. over 30 years. The current bioclimatic variables (bio1–bio19, 1960–1990) necessary for the MaxEnt model were obtained from the WorldClim website (http://www.worldclim.org/ accessed on 10 March 2024) at a 10 min resolution.

In projecting the potential distribution of *A. sarta* globally under future climate change scenarios for 2070, the CLIMEX model leveraged the Special Report on Emissions Scenarios (SRES) A1B and A2 alongside the global circulation model (GCM) CSIRO-Mk3.0 (CS) from the Center for Climate Research. The A1B SRES depicts a future world with a balanced emphasis on all energy sources, encompassing fossil fuels, renewable energy, and nuclear power. Conversely, the A2 SRES provides insights into demographic, technological, and economic variables associated with greenhouse gases (GHG) (da Silva et al., 2017) [44]. The future climate data used for the MaxEnt model comprised the bioclimatic variables (bio1–bio19 CCSM4 (community climate system model) dataset) projected for 2070 under the Representative Concentration Pathway (RCP) 6 and 8.5 scenarios as outlined in the Intergovernmental Panel on Climate Change (IPCC) fifth assessment report (AR5). These datasets are accessible via the CMIP5 and WorldClim website (http://www.worldclim.org/ accessed on 11 March 2024). RCPs are categorized into four types (RCP-2.6, RCP-4.5, RCP-6.0, and RCP-8.5), delineated by their approximate total radiative forcing in 2100, with higher values indicating more severe warming [45].

In terms of CO_2_ concentration projections by the end of this century, the A1B SRES assumes an increase to 720 ppm (parts per million) compared to 670 ppm (RCP-6), while the A2 SRES assumes a higher concentration of 846 ppm compared to 936 ppm (RCP-8.5). Regarding the predicted temperature rise associated with these scenarios, the A1B SRES forecasts an approximate temperature increase of around 4 °C compared to RCP-6 (3 °C), while the A2 SRES anticipates a higher temperature rise of about 6 °C compared to RCP-8.5 (5 °C) [45,46]. The prediction results for future climate indicate that the A1B SRES scenario aligns closely with Representative Concentration Pathway RCP-6 [6], while the A2 SRES scenario corresponds to RCP-8.5 [31]. Due to the absence of RCP scenario data in the CliMond database, we opted for these two future scenarios as the basis for climate variables in CLIMEX models under projected future climate conditions.

### 2.3. Species Distribution Models (SDMs)

The ‘Compare Location’ function within CLIMEX version 4.0 (Hearne Scientific Software, Australia) was employed to conduct simulation analysis. CLIMEX utilizes biological parameters and climate variables to predict the suitable distribution area of a species, utilizing a combination of inductive and deductive methods for fitting, as outlined by Finch et al. [47]. The model evaluates the climatic optimum for the target species at the designated location by computing an ecoclimatic index (EI), which spans from 0 (indicating unsuitability) to 100 (representing suitability year-round). Given the dynamic nature of climate across various regions, achieving a perfect score of 100 is challenging. The EI value is derived not only from the stress index (SI) and annual growth index (GI) but also from the stress interaction index (SX), contributing to a more comprehensive calculation formula: EI = SI × GI × SX [37]. We compiled all parameters in Table 1 and assessed their sensitivity. Appendix A provides a comprehensive overview of the parameter estimation process outlined by Hayat et al. [6]. To offer a detailed description of the favorability of various regions for *A. sarta*, we categorized the ecoclimatic index (EI) values into four distinct groups: unsuitable habitats (EI = 0), marginal suitable habitats (0 < EI ≤ 15), suitable habitats (15 < EI ≤ 30), and highly suitable habitats (EI > 30) [6].

MaxEnt 3.4.4 (available at https://biodiversityinformatics.amnh.org/open_source/maxent/ accessed on 12 March 2024) was utilized to forecast the global distribution range of *A. sarta* across varied climatic conditions. MaxEnt functions by discerning correlations between documented occurrence points and background data within specified constraints, as detailed by [48]. Renowned for its capacity to incorporate diverse variables like climatic data, offer multiple metrics for model performance evaluation, and provide user-friendly software operation, MaxEnt has been widely employed in predicting species distribution [49]. A comprehensive set of 19 bioclimatic variables was acquired from the WorldClim website (http://www.worldclim.org/ accessed on 12 March 2024) and initially employed as potential predictors (Table 2).

The best variables for the final model were identified using the following methods to mitigate multicollinearity among correlated and redundant environmental variables. Initially, the percentage contribution values of the 19 variables to the model were computed by establishing an initial model. The existing records of *A. sarta* and the 19 environmental variables were uploaded to the initial model, with the ‘random test percentage’ set at 25%. Jackknife analysis assessed variable importance, while default values were retained for the remaining model parameters.

In the second step, attribute values of the 19 variables at each of the 51 distribution points were extracted utilizing ArcGIS 10.7 (ESRI, Redlands, CA, USA). Subsequently, all values were imported into IBM SPSS Statistics 26 to compute Pearson correlation coefficients between any two variables. Finally, variables underwent screening: if the correlation coefficient exceeded 0.8, the variable with the highest contribution value was preserved in the initial model, while another variable was omitted [48]. Six environmental factors (out of 19) were selected for inclusion in the final model after screening (Figure 1).

To build the final model, we optimized the Regularization Multiplier (RM) and Feature Class (FC) parameters in the R 3.6.3 software using the Kuenm (https://github.com/marlonecobos/kuenm accessed on 24 April 2024) package [50]. The model can be smoothed out and its overfitting reduced by optimizing these two parameters. The model was chosen with consideration for three key factors: complexity (AICc values), predictive ability (low omission rates), and statistical significance (partial ROC). Following the selection of statistically significant candidate models, the set of models was further narrowed down based on their missing rate (<5%). Ultimately, final models were determined by selecting those with delta AICc values of ≤2.

The final model was executed using the 10-fold bootstrapping crossover method for current and future conditions. Model performance was assessed using the average area under the receiver operating characteristic curve analysis (AUC). Typically, AUC values below 0.6 signify unqualified performance, 0.6 to 0.7 indicate poor performance, 0.7 to 0.8 indicate medium performance, 0.8 to 0.9 indicate good performance, and values above 0.9 indicate high performance [51]. The model simulations were visualized and reclassified using ArcGIS 10.7 software. The potential distribution of *A. sarta* was mapped based on the probability (*p*) of occurrence, ranging from 0 to 1. The Maximum Test Sensitivity Plus Specificity (MTSPS) threshold was utilized to reclassify the probability (*p*) into four categories: unsuitable habitats (*p* ≤ MTSPS), marginal suitable habitats (MTSPS < *p* ≤ 0.35), suitable habitats (0.35 < *p* ≤ 0.65), and highly suitable habitats (*p* > 0.65) [31,48].

### 2.4. Creation of Combined Distribution Maps

Combination maps of two models for two periods were generated using the ‘spatial analysis tools’ functions in ArcGIS 10.7. Climate-suitable regions were identified using a binary suitability threshold: areas where the outputs of CLIMEX had EI > 0 and the outputs of MaxEnt had *p* > 0.341 were selected.

## 3. Results

### 3.1. Model Performance

Based on the Kuenm package’s optimization results, the feature classes (FC) with threshold (T), product (P), quadratic (Q), and regularization multiplier (RM) equal to 1.9 were the optimal MaxEnt model parameters for our data (Table 3). The test AUC values from the 10-fold cross-validation of the final model are depicted in Figure 2. The AUC values were 0.95, signifying the strong performance of the MaxEnt model in predicting the potential distribution areas of *A. sarta*. Under current climatic conditions, the potential distribution area forecasted by the CLIMEX model encompassed all recorded known distributions of *A. sarta* globally. Similarly, the prediction results of MaxEnt indicated that all occurrence records fell within the predicted range. These findings suggest that the prediction results of both models demonstrated good accuracy.

### 3.2. Potential Distribution of A. sarta under Current Climatic Conditions Using CLIMEX and MaxEnt

Under current climatic conditions, the CLIMEX model forecasts that the potential distribution areas predominantly encompass central and southern countries across the world map, with central African, southern Asian, and most northern hemisphere countries deemed unsuitable (Figure 3). The projected proportions for highly suitable, suitable, marginally suitable, and unsuitable areas are 27.07% (36.41 million km^2^), 9.15% (12.30 million km^2^), 4.84% (6.51 million km^2^), and 58.94% (79.28 million km^2^), respectively, relative to the total global land area (Appendix A).

Conversely, the outcomes generated by the MaxEnt model demonstrate a more conservative distribution, primarily concentrated in the central regions of the world. Only three countries—Australia, Chile, and Argentina—in the southern hemisphere are deemed suitable, while the rest of the world is predicted to be unsuitable (Figure 3). The projected proportions for highly suitable, suitable, marginally suitable, and unsuitable areas are 1.95% (2.62 million km^2^), 2.48% (3.34 million km^2^), 6.39% (8.59 million km^2^), and 89.18% (119.94 million km^2^), respectively, relative to the total global land area (Appendix A).

### 3.3. Potential Distribution of A. sarta under Future Climatic Conditions Using CLIMEX and MaxEnt

The CLIMEX model forecasts that, under future climatic conditions in both climate change scenarios, potential distribution areas will primarily be located in the central and southern parts of the world. However, the range of suitable habitats is projected to shift towards the northern hemisphere and contract towards the southern hemisphere. Nonetheless, countries with known potential distribution areas of *A. sarta* will likely remain suitable habitats for this pest (Figure 4). Regarding area distribution under the SSP370 and SSP585 climate change scenarios, a notable increase has been observed in suitable habitat (0.29 and 0.34 million km^2^), marginal suitable habitat (0.32 and 0.51 million km^2^), and unsuitable habitat (1.27 and 1.22 million km^2^). However, the highly suitable habitat area under both climate change scenarios is anticipated to decrease significantly by −1.88 and −2.06 million km^2^, respectively (Appendix A).

The MaxEnt model forecasts that, under future climatic conditions in both climate change scenarios, potential distribution areas will primarily be located in the central parts of the world. However, the range of suitable habitats is projected to shift towards the northern hemisphere, with significant changes expected in Asia, Europe, and North America, while contracting towards the southern hemisphere. Despite these shifts, countries with known potential distribution areas of *A. sarta* will likely remain suitable habitats for this pest (Figure 4). Regarding area distribution under the SSP370 and SSP585 climate change scenarios, a notable increase has been observed in highly suitable habitat (0.30 and 0.23 million km^2^), suitable habitat (0.82 and 0.66 million km^2^), and marginal suitable habitat (3.73 and 2.34 million km^2^). However, the unsuitable habitat area under both climate change scenarios is anticipated to decrease significantly by −4.84 and −3.23 million km^2^, respectively (Appendix A).

### 3.4. Net Change in Aeolesthes sarta Global Distribution under Future Climate Using CLIMEX and MaxEnt

The net change in *Aeolesthes sarta*’s global distribution under future climate scenarios differs between the CLIMEX and MaxEnt models. In the CLIMEX model, net losses predominantly occur in the central regions, shifting toward the southern hemisphere, while net gains predominantly occur towards the northern hemisphere. Conversely, the MaxEnt model indicates net losses primarily concentrated in the southern hemisphere and net gains mainly in the central regions of the world. CLIMEX forecasts reveal a contraction of suitable habitats from the center towards the northern hemisphere. Meanwhile, MaxEnt projections highlight a slight shift in suitable habitats towards the northern hemisphere but predominantly in the central regions, particularly Asia, Europe, and North America (Figure 5).

### 3.5. Combined Prediction Maps of the Two Models

The combined distribution maps (Figure 6) illustrate that, under current and future climate conditions, the intersected climate-suitable regions of the CLIMEX and MaxEnt models predominantly encompass the central and southern countries of the world. Notably, the results of the CLIMEX projections include all those from the MaxEnt model. With anticipated future climate changes, the intersecting areas shift primarily northward and are concentrated in Asia, Europe, and North America, indicating a significant alteration in the distribution patterns of suitable habitats.

Under historical and future climate conditions, the output results (EI) of the CLIMEX model demonstrated heightened sensitivity to changes in DV0 (limiting low temperature °C), suggesting that low temperatures significantly influence the distribution of *A. sarta*. Conversely, the MaxEnt model highlighted the significance of Bio3 (isothermality) in model construction, underscoring the importance of temperature dynamics (Figure 7). The CLIMEX model predicts a broader global distribution of *A. sarta* compared to MaxEnt.

### 3.6. Effect of Environmental Factors

At the global level, the top environmental variables associated with *A. sarta* distribution were ranked as follows: isothermality, temperature seasonality, mean annual temperature, and precipitation seasonality. These variables made average contributions of 33.1%, 24.5%, 21.7%, and 17% to the model, respectively (Table 4). The jackknife tests of variable importance further confirmed that the variables isothermality, temperature seasonality, mean annual temperature, and precipitation seasonality exhibited higher predictive power than others. This was evident from their high training, test gain, and AUC values, indicating their significant contribution to the model’s predictive accuracy (Figure 8).

The likelihood of *A. sarta* presence exhibits distinct trends based on mean annual temperature. It sharply increases from 0 to 13 °C, peaks at 14 °C, and declines rapidly beyond 15 °C (Figure 9A). Additionally, areas with moderate temperature uniformity throughout the year, where seasonal changes contribute significantly to annual temperature variation, show higher probabilities of *A. sarta* presence (Figure 9B). Similarly, regions characterized by consistent temperatures year-round and minimal seasonal fluctuations display increased probabilities of *A. sarta* presence (Figure 9C). Furthermore, the probability peaks between −10 °C and 17 °C, with the highest likelihood observed at 18 °C, diminishing sharply after 20 °C (Figure 9D). Areas with significant precipitation variability, indicative of distinct wet and dry seasons or irregular rainfall patterns, exhibit higher probabilities of *A. sarta* presence (Figure 9E). Lastly, regions experiencing low precipitation during the coldest quarter of the year also show elevated probabilities of *A. sarta* presence (Figure 9F).

The CLIMEX model forecasts a greater growth index in regions where the *A. sarta* is observed. This outcome is attributed to the heat and cold stress indexes, which elucidate why *A. sarta* avoids extremely cold northern latitudes and hot desert regions in Africa and certain parts of Australia (Figure 10).

## 4. Discussion and Conclusions

In this research, two models were employed to forecast the potential distribution range of *A. sarta* worldwide, thereby mitigating the limitations of relying on a single model. The CLIMEX model specifically examines the correlation between species distribution and environmental factors, incorporating ecophysiological data that influence population growth rates across four stress indicators: hot, cold, dry, and wet [52]. CLIMEX stands out for its ability to forecast invasive species even with limited distribution data, leveraging biological characteristics of the species alongside climatic information [6]. This feature constitutes a primary advantage of the model. Moreover, CLIMEX offers simplicity in operation and allows for fine-tuning predicted distribution ranges based on actual data. It can also be updated to reflect changes in invasive species’ biological parameters, thereby enhancing the accuracy of predictions. However, defining the relationship between biological parameters and growth rates is a user-driven task, introducing subjectivity into the simulation results and making them susceptible to the modeler’s biases [6,7]. The mechanistic physiologically based approach of CLIMEX, along with its independence from presence background data, has resulted in a dearth of corresponding evaluation metrics [32].

In contrast, MaxEnt primarily relies on independent data concerning species distribution and environmental variables for its predictions. Its key advantage lies in its ability to incorporate a wider array of environmental variables, including soil and elevation factors, without necessitating comprehensive knowledge of the species’ biological characteristics. However, the model’s accuracy may be influenced by the number of species distribution sites available [53]. The relative occurrence rate (ROR), as defined by Fithian & Hastie [54], corresponds to the raw output generated by the MaxEnt model. Predicted ROR values are primarily depicted by creating intricate, highly nonlinear response curves derived from the model’s formulas. This process underscores that the model integrates both statistical and machine learning approaches. The selection of environmental characteristics, discretization, and sample size significantly influence the prediction outcomes. Consequently, there is a risk that the model may diverge from macroecological patterns [55]. Early et al. [56] emphasize that using species range projections for prevention strategies warrants using multiple modeling techniques. These techniques should rely on independent or semi-independent data regarding the known distribution of species. Moreover, the output results from different models should be formally comparable. Consequently, combining the prediction results from various models can enhance the accuracy of range predictions.

Our study addresses the pressing need for accurate models predicting the potential distribution of economically significant pests, which is crucial for conducting pest risk assessments and is essential for facilitating international trade. We have pioneered the global assessment of potential risk for *A. sarta* establishment, successfully employing correlative niche models like MaxEnt and semi-mechanistic niche models like CLIMEX. Our approach holds promise for assessing other agricultural and forest pests of quarantine concern. The MaxEnt model leveraged *A. sarta* occurrences and a comprehensive set of environmental spatial data layers, while CLIMEX utilized published physiological tolerance data specific to *A. sarta* alongside built-in climate data layers to predict the potential for establishment. Notably, both models accurately predicted *A. sarta*’s known occurrences. However, they did not project suitable environmental conditions in countries situated in extreme northern hemispheres, African deserts, and Australia primarily due to the adverse effects of extreme chilling temperatures or excessively high temperatures in these regions, which hinder *A. sarta* development or diapause regulation. The models have forecasted suitable conditions in several countries where *A. sarta* is not currently present, such as in various European and North American countries, where its preferred host species, *Populus alba*, is found (refer to Figure 11). Among the environmental variables considered, average annual temperature and precipitation rate emerged as the top factors associated with *A. sarta* distribution. These findings underscore the importance of climatic conditions in determining the potential range of *A. sarta* and highlight areas where proactive measures may be necessary to prevent its establishment.

### 4.1. Assessment of CLIMEX and MaxEnt Models

The disparity in prediction results between CLIMEX and MaxEnt can be attributed to their distinct methodologies. CLIMEX primarily relies on the biological parameters of the species, while MaxEnt necessitates specific species distribution sites for modeling. CLIMEX characterizes the relationship between the environment and distribution by incorporating ecophysiological parameters that govern species survival and population growth rates [57]. Following the invasion of *A. sarta* into other Asian countries, changes occurred in the heating and chilling points relative to its native range in the subcontinent. Consequently, CLIMEX parameters were readjusted to extend the predicted distribution of the model to cover all known occurrences. In contrast, the MaxEnt model employs complex, nonlinear functions to fit species responses to the relevant environment. These functions involve transformations applied to independent variables, and the fitted function is defined by six feature classes: linear, product, quadratic, hinge, threshold, and categorical [58]. This complexity accounts for the different outputs and approaches between CLIMEX and MaxEnt. Indeed, several factors can contribute to differences in prediction results among models. These factors include the algorithm used in the model, spatial errors in species occurrences, multicollinearity among environmental variables, and genetic variation among species [59]. The interaction of these elements can influence the accuracy and reliability of model predictions, highlighting the importance of considering various factors and employing multiple modeling techniques when assessing species distribution and potential range expansion.

### 4.2. Global Projections of A. sarta Distribution under Current and Future Scenarios

The predictions generated by both models under current climatic conditions closely align with the currently known global distribution of *A. sarta* and its preferred host plant, *P. alba* (Figure 3 and Figure 11). For instance, both models accurately projected *A. sarta* distribution in countries such as Pakistan, Iran, India, China, Afghanistan, Turkmenistan, Tajikistan, Uzbekistan, and Kazakhstan, consistent with reported occurrences in these regions [10,14,15,25,26,28]. Under future climatic conditions, both models anticipate a highly suitable area concentrated in central regions of the world, with significant extensions toward the northern hemisphere and contractions toward the southern hemisphere (Figure 5). While the CLIMEX model predicts a larger area than MaxEnt, both models’ overall accuracy remains high. This alignment between model predictions and observed occurrences underscores the effectiveness of both CLIMEX and MaxEnt in forecasting the potential distribution of *A. sarta* under changing climatic conditions. The disparities observed in spatial predictions between the CLIMEX and MaxEnt models can be attributed to several factors. These include utilizing different types and spatial resolutions of climatic datasets, variations in the complexity of model fitting, and specific assumptions inherent to each model [58,60].

### 4.3. Caveats and Uncertainties

The findings of this study should be approached with caution due to the inherent uncertainties associated with niche models. Niche model predictions can be influenced by various factors, including the quality of occurrence data, potential sampling bias, resolution of spatial data layers, species characteristics, and spatial autocorrelation [61,62]. It is important to note that the physiological temperature and moisture thresholds derived from laboratory studies for *A. sarta* may not encompass the full spectrum of genetic and phenotypic variability present in *A. sarta* populations worldwide. The CLIMEX parameters used in modeling also introduce uncertainties, as highlighted in the literature [63]. The MaxEnt model is susceptible to variations from diverse decisions made during its calibration process. Choices such as the selection of background points, determination of extent, setting the value of the regularization multiplier (RM), and choosing feature types exert significant influence on the outcomes of the model predictions [64,65]. Our validation of MaxEnt predictions, employing a semi-mechanistic CLIMEX model, indicates that the decisions made during the calibration process for the *A. sarta* MaxEnt model were suitable. This affirmation is supported by the broad alignment between model projections and observations across significant regions (Figure 3). Climate change is anticipated to influence the distribution of *A. sarta*, as projected by the CLIMEX and MaxEnt models. With the rise in global temperatures, regions exhibiting marginal suitability for *A. sarta* (with an average annual temperature of 10 °C) are expected to become more conducive to its presence. Conversely, areas experiencing higher average annual temperatures (>37 °C) will likely become unsuitable for *A. sarta*. Further studies are warranted to explore the potential impacts of climate change on the distribution and biology of *A. sarta*. Temperature and moisture level changes can influence insect pests’ population growth rates, extend the number of generations, prolong the development season, and alter forest–pest synchrony and interspecific interactions [66]. Improving the temporal resolution of climate data may be necessary for more accurate predictive models of insect pest establishment. Monthly averaged climatic data, such as those provided by WorldClim [67], may not adequately capture the critical physiological requirements of certain insect pests, which may necessitate finer resolution data at the weekly or daily level [68]. Beyond the influence of a suitable climate, the probability of an insect pest establishing itself in new geographical areas is also impacted by factors such as propagule pressure (the number of individuals introduced to a novel region), the presence of host plant species, and various abiotic factors and biotic interactions. These interactions include competitors and natural enemies [69,70]. Our models predicted climatic suitability in several regions, where *A. sarta* currently does not occur (e.g., in Europe and North America—while all those regions have well-established *P. alba* plantations), which may be because of very low propagule pressure, dispersal barriers, and natural enemies in these regions [3]. The propagule pressure in different parts of the world depends on the frequency and amount of timber/furniture imports and the likelihood of these wood logs infested with *A. sarta*.

### 4.4. International Trade Implications for Biosecurity

Our findings hold significant implications for guiding pest risk assessments conducted by national plant protection organizations, monitoring efforts to prevent unintentional introductions of *A. sarta* in various countries and informing policymakers and trade negotiators to make science-based decisions. By strategically co-ordinating and concentrating efforts across susceptible areas, we can mitigate the risk of *A. sarta* incursions. Even in regions like Europe and North America, where *A. sarta* currently does not exist, our results can facilitate the implementation of effective monitoring and surveillance programs to detect potential pest introductions via trade from currently infested countries or areas with high climatic suitability. Furthermore, the maps generated from our study can aid in identifying areas most suitable for area-wide pest suppression strategies, including the sterile insect technique or eradication efforts. Targeting regions with established populations of *A. sarta* on the extreme margins of climate suitability can enhance the efficacy of suppression and eradication initiatives. Our research provides valuable insights that can inform proactive and strategic pest management practices globally.

Authors should discuss the results and how they can be interpreted from the perspective of previous studies and of the working hypotheses. The findings and their implications should be discussed in the broadest context possible. Future research directions may also be highlighted.

## Figures and Tables

**Figure 1 insects-15-00324-f001:**
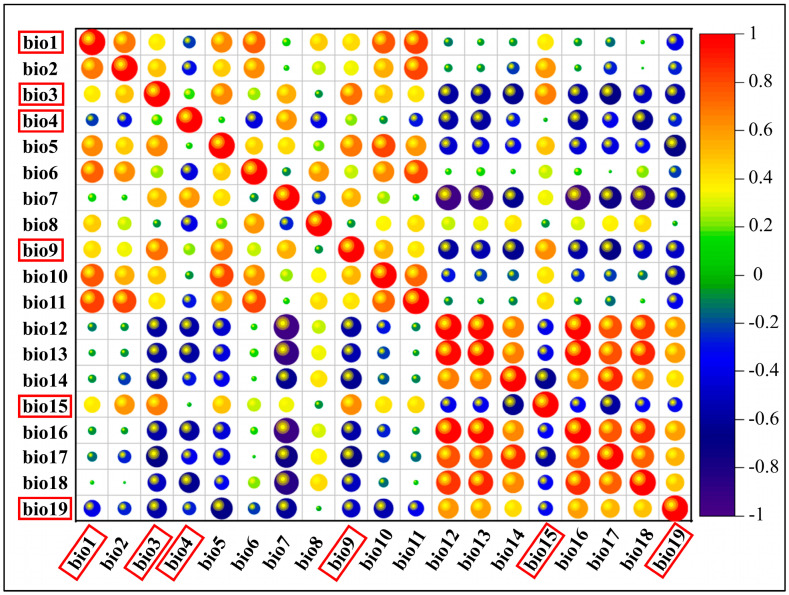
Pearson’s correlation analysis of the environmental variables in the initial model. The variables chosen for constructing the MaxEnt models are indicated with red box.

**Figure 2 insects-15-00324-f002:**
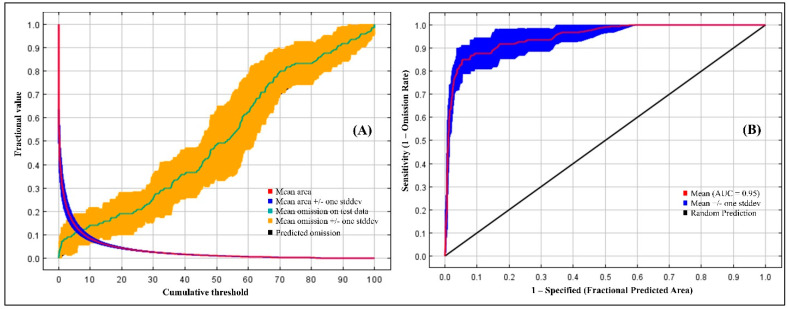
Model evaluations, (**A**) averaged omission and predicted area for *A. sarta*; (**B**) the ROC curve calculated by MaxEnt as averaged sensitivity versus 1-specificity for *A. sarta*.

**Figure 3 insects-15-00324-f003:**
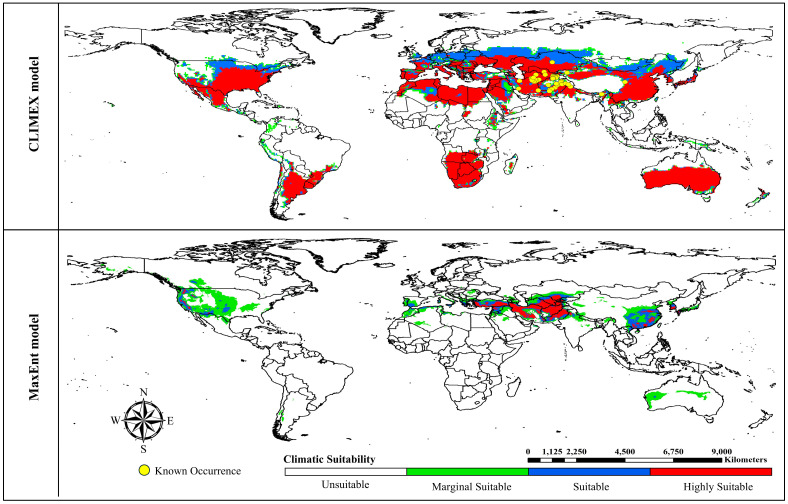
Potential global distribution of *Aeolesthes sarta* under current climate conditions with the CLIMEX and MaxEnt model. Habitat suitability presented with four colors; red = highly suitable, blue = suitable, green = marginal suitable, and white = unsuitable habitat.

**Figure 4 insects-15-00324-f004:**
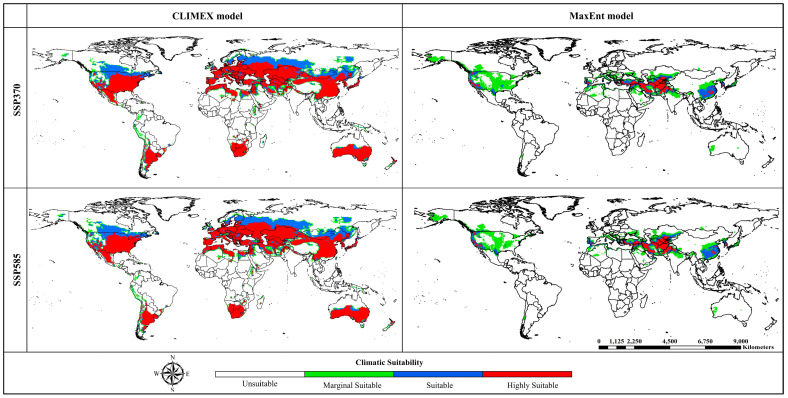
Potential global distribution of *Aeolesthes sarta* under future climatic conditions for two climate change scenarios with the CLIMEX and MaxEnt model. Habitat suitability presented with four colors; red = highly suitable, blue = suitable, green = marginal suitable, and white = unsuitable habitat.

**Figure 5 insects-15-00324-f005:**
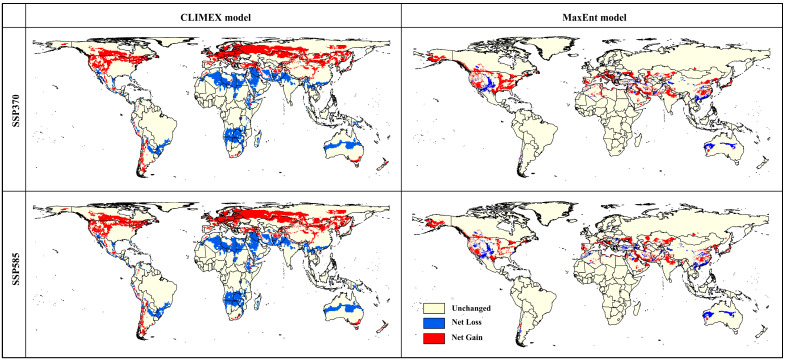
Net gain and loss of *Aeolesthes sarta* potential distribution under future climate for CLIMEX and MaxEnt models.

**Figure 6 insects-15-00324-f006:**
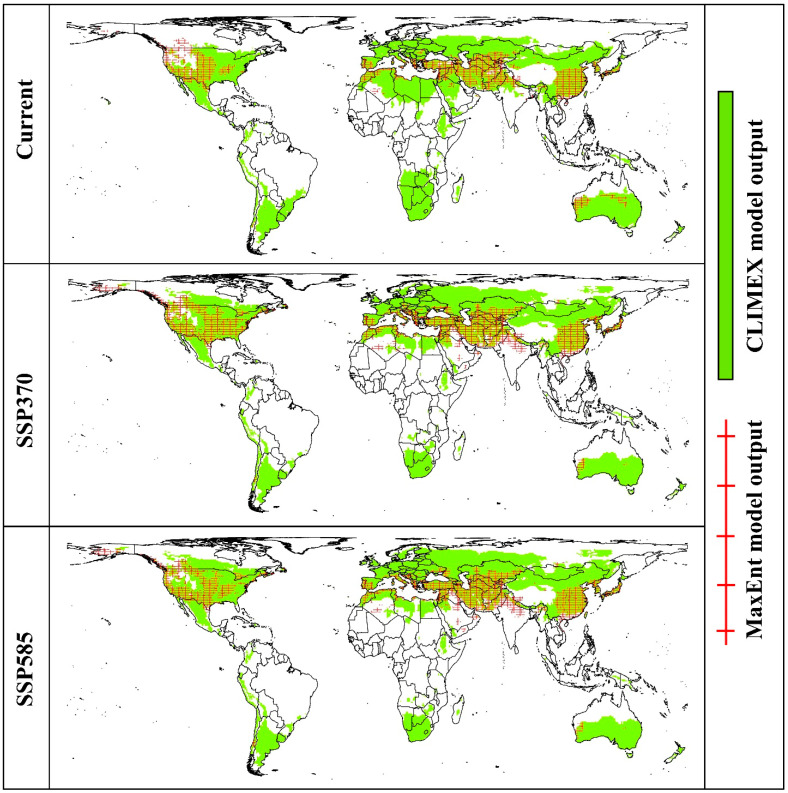
Combined prediction maps intersected with the projected suitable habitats from the CLIMEX and MaxEnt models under current and future climate conditions.

**Figure 7 insects-15-00324-f007:**
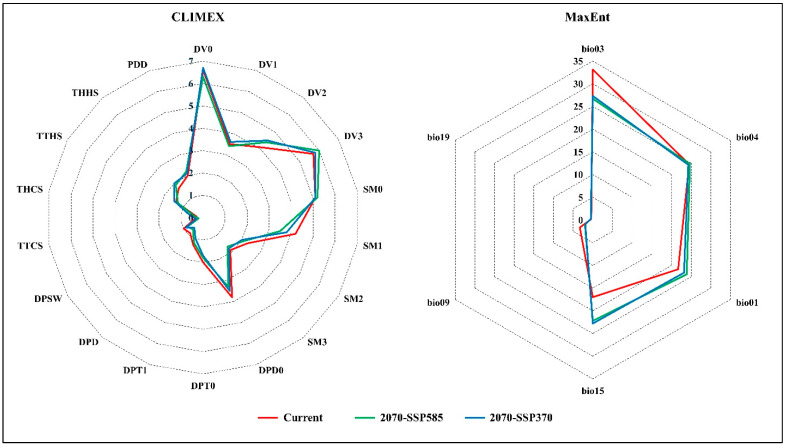
The comparisons of the CLIMEX sensitivity of ecophysiological parameters and the MaxEnt variable contribution.

**Figure 8 insects-15-00324-f008:**
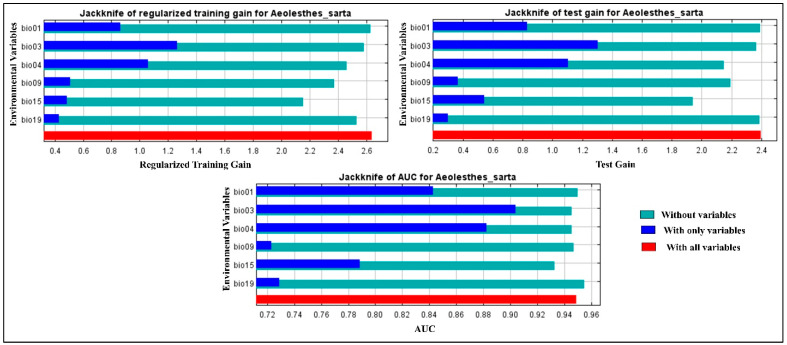
The relative importance of the environmental variables based on the jackknife test. The figures show each variable’s contribution to regularized training gain, test gain, and AUC in the *A. sarta* model.

**Figure 9 insects-15-00324-f009:**
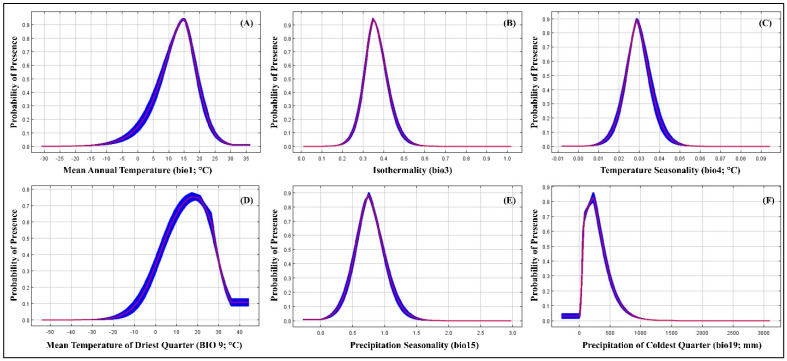
Response curves of the best predictors of *A. sarta* in the MaxEnt model: (**A**) mean annual temperature (bio1; °C), (**B**) isothermality (bio3), (**C**) temperature seasonality (bio4), (**D**) mean temperature of the driest quarter (bio9; °C), (**E**) precipitation seasonality (bio15), and (**F**) precipitation of coldest quarter (bio19; mm).

**Figure 10 insects-15-00324-f010:**
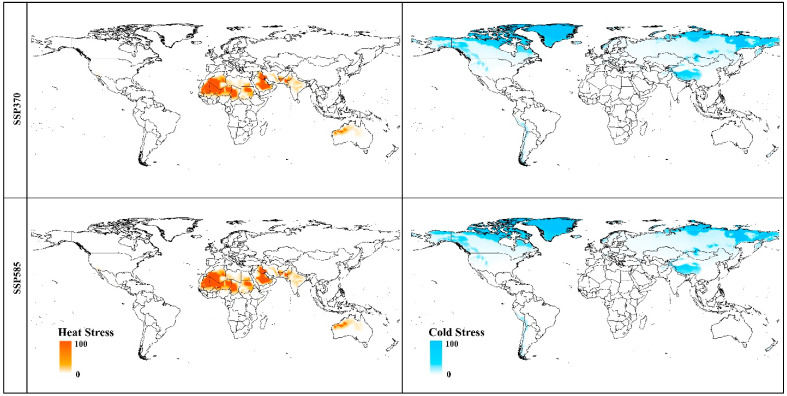
The predicted region where *A. sarta* survival will be affected globally by cold stress and heat stress under climate change scenarios in 2070.

**Figure 11 insects-15-00324-f011:**
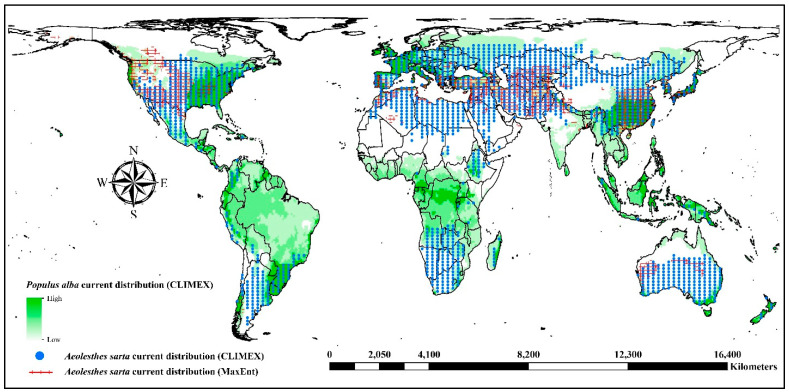
Current climatic suitability for the growth of *Populus alba* at a global scale modeled by CLIMEX. Most countries with suitable environmental conditions have a known distribution of *P. alba*. Please refer to Hayat et al. [6] for details.

**Table 1 insects-15-00324-t001:** Parameter values used in the CLIMEX model for *Aelosthes sarta*.

Parameters		Code	Values
			*Aeolesthes sarta*
**Temperature**	Limiting low temperature (°C)	DV0	10
	Lower optimal temperature (°C)	DV1	15
	Upper optimal temperature (°C)	DV2	37
	Limiting high temperature (°C)	DV3	40
**Moisture Index**	Limiting low soil moisture	SM0	0
	Lower optimal soil moisture	SM1	0.001
	Upper optimal soil moisture	SM2	1.5
	Limiting high soil moisture	SM3	2.5
**Diapause Index**	Diapause induction day length	DPD0	12
	Diapause induction temperature (°C)	DPT0	13
	Diapause termination temperature (°C)	DPT1	10
	Diapause development days	DPD	90
	Summer or winter Diapause	DPSW	0
**Cold Stress**	CS temperature threshold (°C)	TTCS	9
	CS temperature rate	THCS	−0.00001
**Heat Stress**	HS temperature threshold (°C)	TTHS	41
	HS temperature rate	THHS	0.005
**Population degree day**		PDD	700

Source: [6,7].

**Table 2 insects-15-00324-t002:** List of environmental variables used in MaxEnt modeling.

Environmental Variable	Interpretation
bio1	Annual mean temperature
bio2	Mean diurnal range (mean of monthly (max temp − min temp))
bio3	Isothermality (Bio2/Bio7) (*100)
bio4	Temperature Seasonality (standard deviation * 100)
bio5	Max Temperature of Warmest Month
bio6	Min Temperature of Coldest Month
bio7	Temperature Annual Range (Bio5–Bio6)
bio8	Mean Temperature of Wettest Quarter
bio9	Mean Temperature of Driest Quarter
bio10	Mean Temperature of Warmest Quarter
bio11	Mean Temperature of Coldest Quarter
bio12	Annual precipitation Seasonality
bio13	Precipitation of Wettest Month
bio14	Precipitation of driest month
bio15	Precipitation Seasonality (Coefficient of Variation)
bio16	Precipitation of Wettest Quarter
bio17	Precipitation of Driest Quarter
bio18	Precipitation of Warmest Quarter
bio19	Precipitation of Coldest Quarter

Source: http://www.worldclim.org/ (accessed on 12 March 2024).

**Table 3 insects-15-00324-t003:** The best candidate models filtered by the Kuenm package.

Rank	FC	RM	Partial ROC	Omission Rate at 5%	AICc	Delta AICc
1	QPT	1.9	0	0.046	9508.12	0

**Table 4 insects-15-00324-t004:** The bioclimatic variables percentage contribution and permutation in the final MaxEnt model.

Variable	Percentage Contribution	Permutation Importance
bio03	33.1	25.4
bio04	24.5	34.6
bio01	21.7	15.8
bio15	17	6.4
bio09	3.3	16.5
bio19	0.4	1.3

## Data Availability

Data will be available upon request.

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
