# Peer review of "Which SDM Model, CLIMEX vs. MaxEnt, Best Forecasts Aeolesthes sarta Distribution at a Global Scale under Climate Change Scenarios?"

_insects, 2024, doi:10.3390/insects15050324_

Round 1
Reviewer 1 Report
Comments and Suggestions for Authors
Generally a well-written comparison of invasive pest models. The information presented was relevant, but the introduction lacked a cohesive flow and felt redundant at times, so could be improved for clarity. The results were described in sufficient detail, but textual statements did not always align with visual representations. Figure quality could also be improved as many were blurry and difficult to read. Conclusions and caveats were appropriately discussed. See attached document for detailed comments.

Author Response
All the comments has been addressed, for details please refer to the MS.
Reviewer 2 Report
Comments and Suggestions for Authors
L 33 – Italicize genera
L37 – Best practice to use maximum entropy anytime you are talking about the models and modelling technique, and only MaxEnt went referring to the software/package.
L 72 – Since Aeolesthes sarta is the more commonly used name in the literature, I suggest the sentence be “Aeolesthes sarta (syn. Trirachys sartus), commonly known as SLB (Sarta Longhorned Beetle), is a prominent member of Cerambycidae” – you call it Aeolesthes sarta but then say it’s in the genus Trirachys, which then is contradictory.
L 143 – Research question asks which is more accurate, but there is no test for accuracy with a testing data set. While goodness of fit for the models can be assessed with AUC, that does not provide any quantification of accuracy or distinguish which model produces the most accurate prediction. This can only come from a separate testing data set.
L 274 – Provide justification as to why these thresholds (EI > 0 and p > 0.3) were selected. For a maximum entropy model, 0.3 seems extremely low, especially without a justification.
Figure 3 – Throughout the paper, the order has been CLIMEX then MaxEnt. Flip the arrangement so CLIMEX is first/on-top. Also, the color legend needs further clarification to define the how/thresholds/etc. that determine the categories (Unsuitable, Marginal Suitable, Suitable, Highly Suitable).
Figure 4 – Same issue as with Figure 3, no definition as to the color categories in the legend. Upon further inspection Figure 4 is not needed – there is no real visible change with the predicted climate conditions. The total area does not change, so why include the figure? The text in L 316-326 captures this for CLIMEX – the largest change predicted was in the highly suitable area and that was 2.06 million km² (1.4% of land). Since the numbers are presented in text, I do not believe Figure 4 is even necessary to visualize a 1% change. Same for MaxEnt, the authors provide the numbers in text.
Figure 5 – Again, since the order has been CLIMEX then MaxEnt, CLIMEX should be on the left (same for Figure 6). Again, the colors are undefined and there is no information as to thresholds for the different categories provided.
Figure 12 – I’m of the opinion that you cannot introduce new information into the discussion. The figure for Populus alba should be presented in the results as a combination of the model outputs and the preferred host suitable climate. Then it can be interpreted as part of the discussion.
L 454 – Yes, accurate models are needed, however, the authors did not assess accuracy in either model. As such, there is still a potential need for an accurate model.
L 465 – The authors are interpreting beyond their data. As there was no assessment of accuracy, It is in bad form to suggest that the “models accurately predicted A. sarta’s known occurrences.” In fact, the ROC curves only provide confidence that the models provided a good-fit to the data, not that they could predict occurrences from a test data set.
Author Response
Thank you for taking the time to review our MS. We have tried our best to address all the questions and suggestions you have made. Please find the details in attach file below and MS.
